# Subtypes of Adult-Onset Asthma at the Time of Diagnosis: A Latent Class Analysis

**DOI:** 10.3390/ijerph20043072

**Published:** 2023-02-09

**Authors:** Elina M. S. Mäkikyrö, Maritta S. Jaakkola, Taina K. Lajunen, L. Pekka Malmberg, Jouni J. K. Jaakkola

**Affiliations:** 1Center for Environmental and Respiratory Health Research, Research Unit of Population Health, University of Oulu, FI-90014 Oulu, Finland; 2Biocenter Oulu, University of Oulu, FI-90014 Oulu, Finland; 3Medical Research Center Oulu, Oulu University Hospital, FI-90220 Oulu, Finland; 4Skin and Allergy Hospital, Helsinki University Hospital, University of Helsinki, FI-00280 Helsinki, Finland

**Keywords:** adult-onset asthma, subtype, phenotype, risk-factors of asthma subtypes

## Abstract

Introduction: Only a few previous studies have investigated the subtypes of adult-onset asthma. No previous study has assessed whether these subtypes are different between men and women, or whether these subtypes have different risk factors. Methods: We applied latent class analyses to the Finnish Environment and Asthma Study population, including 520 new cases of adult-onset asthma. We formed subtypes separately between women and men and analyzed the following determinants as potential predictors for these subtypes: age, body mass index, smoking, and parental asthma. Results: Among women, the subtypes identified were: 1. *Moderate asthma*, 2. *Cough-variant asthma*, 3. *Eosinophilic asthma*, 4. *Allergic asthma*, and 5. *Difficult asthma*. Among men, the subtypes were: 1. *Mild asthma*, 2. *Moderate asthma*, 3. *Allergic asthma*, and 4. *Difficult asthma*. Three of the subtypes were similar among women and men: *Moderate, Allergic*, and *Difficult asthma*. In addition, women had two distinct subtypes: *Cough-variant asthma*, and *Eosinophilic asthma*. These subtypes had different risk factor profiles, e.g., heredity was important for *Eosinophilic* and *Allergic asthma* (RR for Both parents having asthma in Eosinophilic 3.55 (1.09 to 11.62)). Furthermore, smoking increased the risk of *Moderate asthma* among women (RR for former smoking 2.21 (1.19 to 4.11)) and *Difficult asthma* among men but had little influence on *Allergic* or *Cough-variant asthma.* Conclusion: This is an original investigation of the subtypes of adult-onset asthma identified at the time of diagnosis. These subtypes differ between women and men, and these subtypes have different risk factor profiles. These findings have both clinical and public health importance for the etiology, prognosis, and treatment of adult-onset asthma.

## 1. Introduction

Asthma is a heterogeneous disease which includes many different subtypes, and the age of onset seems to be a significant factor separating the types of asthma [1,2]. Among a growing number of cluster analyses, only two previous studies focused on subtyping adult-onset asthma [3,4]. It has been suggested that there is an important difference in the etiology and prognosis of adult-onset asthma compared with those of childhood asthma: environmental factors, occupational exposures and smoking play a more pronounced role for the onset of adult-onset asthma, although it also has a hereditary component [5,6,7,8].

A previous review article suggested at least five subtypes of adult-onset asthma based on both previous studies on adult-onset asthma populations and the general adult asthma population [9]. The two previous cluster analyses on adult-onset asthma population reported three and five subtypes [3,4].

Neither of the previous studies on adult-onset asthma analyzed the subtypes separately for men and women. However, differences between men and women are likely to exist, since there is some evidence of such differences from studies on adult asthma patients [10,11]. Sood and co-workers reported in 2013 that adult-onset asthma becomes the dominant asthma type in women around the age of 40 years, whereas among men childhood-onset asthma continues to be more common, even in adulthood [12]. Furthermore, the risk factors for the different subtypes of adult-onset asthma have not been previously studied. Also, to our knowledge, there are no previous studies where cluster analyses have been conducted on treatment-naïve asthma-patients.

We applied latent class analysis on 520 cases of newly diagnosed adult-onset asthma from the population-based Finnish Environment and Asthma Study (FEAS) to identify clinically meaningful subtypes of adult-onset asthma. This will produce information that is useful for more personalized management of asthma and can provide information useful for assessing the prognosis of asthma. We included only variables of asthma manifestation in the classification of asthma subtypes. Furthermore, we studied gender-specific subtypes of adult-onset asthma and investigated risk factor profiles for these identified subtypes.

## 2. Methods

### 2.1. Study Design

The Finnish Environment and Asthma Study (FEAS) is a population-based incident case-control study [13,14]. All new cases of asthma in the Pirkanmaa Hospital District in Southern Finland diagnosed in 1997-2000 among adults aged 21 to 63 years were recruited. Patients were recruited by the physician in all of the healthcare facilities diagnosing asthma in the region and via the National Social Insurance Institution of Finland. Then, controls from working-aged adults in the same geographical area were randomly selected. In this study, the controls were used for the spirometry reference values, but all analyses were conducted on the asthma cases. A total of 362 cases (response rate 90%) participated through the health care system, and 159 cases through the National Social Insurance Institution (response rate 78%), constituting a total of 521 cases. One case was later excluded due to a different ethnic background, which would require different reference values for lung function. The current study includes all of the 520 cases originally included in the study.

The cases were diagnosed to have asthma according to the recommendations by the National Asthma Program in Finland at the time [15]. For asthma diagnosis, the following was required: (a) history of at least one asthma-related symptom and (b) demonstration of reversibility of airway obstruction in lung function parameters. In addition to the lung-function measurements, the data collection included a self-administered questionnaire with sections on personal characteristics, health information, active smoking and environmental tobacco smoke exposure, occupation and work environment, home environment and dietary habits. For the cases who were recruited through the medical institutions, the same diagnostic protocol was applied: spirometry accompanied by a bronchodilation test and a two-week diurnal peak expiratory flow (PEF) follow-up. Additional measurements, e.g., a two-week oral steroid treatment test, were conducted at need if the diagnosis was not confirmed. For those recruited via the National Social Insurance Institution, the data were obtained from the medical records. The data collection and principles of asthma diagnosis have been reported in detail elsewhere [13].

### 2.2. Variables Included in the Latent Class Analyses

We focused on variables that were available in clinical practice at the time of diagnosis. Since latent class analysis assumes independence of variables, we included only single variables to describe any specific clinical feature of the disease [16,17].

Spirometry was recorded before and after bronchodilation medication with a pneumotachograph-type disposable flow transducer (Medikro 905; Medikro, Kuopio, Finland) according to the standards of the American Thoracic Society [18]. The non-smoking controls in the FEAS-study served as the control population and the z-values for the spirometry results were based on their distribution. Age and height were accounted for in the reference calculations.

To determine the type of ventilatory dysfunction in the spirometry, we formed the following combinations of variables by using the prebronchodilator values of FEV1/FVC-ratio (FEV1 = forced expiratory volume in one second, FVC = forced vital capacity) and FVC, expressed as z-values: (1) No obstruction or restriction, when FEV1/FVC and FVC z-values ≥ −1.65, (2) Obstruction, when FEV1/FVC < −1.65 and FVC ≥ −1.65, (3) Restriction, when FEV1/FVC ≥ −1.65 and FVC < −1.65, and (4) Obstruction and restriction, when both FEV1/FVC and FVC < −1.65. The FEV1 result from the spirometry was applied to capture the severity of obstruction among the cases: (1) Very mild (z-value > 1.65), (2) Mild (−1.65–−1.99), (3) Moderate (−2.00–−2.49), (4) Severe (−2.50–−3.99), and (5) Very Severe (≤−4.00).

We used the variable forced expiratory flow at 50% of FVC (FEF50%) to reflect small airway dysfunction: (1) no (z-value > −1.65), (2) yes (≤−1.65).

To illustrate variability of the airflow obstruction, we combined the obstructive findings in the diurnal follow-up of peak expiratory flow (PEF) and bronchodilator responsiveness in spirometry: daily variation ≥ 20% and/or ≥15% improvement in response to short-acting bronchodilation medication in PEF-measurements at least twice for two weeks and/or ≥12% and ≥200 mL improvement in FEV1 or FVC in spirometry in response to bronchodilation medication.

Blood eosinophil level was included as a two-category variable: (1) <0.25 × 109/L, and (2) ≥0.25 × 109/L [19].

Specific IgE antibodies to common aeroallergens, including birch, timothy grass, mugwort, cat, dog, horse, Dermatophagoides pteronyssinus and Aspergillus fumigatus were measured using an immunoassay system Pharmacia UniCAP (Phadia Ab, Uppsala, Sweden; www.phadia.com, accessed on 29 October 2022) for determining the Phadiatop score [20]. Atopy was included applying a three-category ordinal-scale variable: (1) No positive result, (2) One positive result (Sensitized), and (3) Multiple positive results (Polysensitised). A positive result was defined based on the limit of ≥0.35 kU/L.

The result of the methacholine challenge was included as a marker for bronchial hyperresponsiveness (BHR): (1) No = PD20FEV1 > 2600 µg, (2) Mild = 601–2600, (3) Moderate = 151–600, and (4) Marked ≤ 150 [21]. The PD20 indicated the provocative dose of inhaled methacholine which caused a 20% decrease in the FEV1-level.

The following symptoms, experienced during the previous 12 months, were recorded with a standardized questionnaire that the study subjects filled in at their first visit: (1) Persistent cough for over two months, (2) Nocturnal cough, wheezing or shortness of breath, (3) Occasional wheezing of breath, and/or (4) Exercise-induced shortness of breath. The symptoms were recorded as yes or no, and potential missing information was inquired by the research nurse in the first clinic visit.

Asthma-COPD-overlap-syndrome (ACOS) was defined on post-bronchodilation FEV1/FVC-ratio < 0.7. It was present in only 25 subjects (6.4%) [22].

The frequencies of the variables used in the latent class analysis are listed in Table 1 for the whole population and separately for men and women.

### 2.3. Statistical Methods

We applied latent class analysis (LCA) to form the subtypes of Adult-Onset asthma using PROC LCA add-on in SAS statistical software (SAS, version 9.4, SAS Institute, Cary, NC, USA) [16,23]. We conducted a principal component analysis on the symptom variables and noticed that there were two different components formed by the symptoms: one included nocturnal symptoms and prolonged cough, while the other included exercise-induced symptoms and wheezing [24]. Because the symptoms represented similar entities, we selected occasional wheezing of breath and prolonged cough to represent these components in the analyses. We calculated the risk factors predicting each subtype of asthma by allocating the subjects in their best fitting latent class. We then conducted multivariate Poisson regression analyses using the PROC GENMOD–procedure. The other classes were combined to form the reference category. The covariates fitted were age [25], smoking [14], BMI [26], and parental asthma [8].

We identified the best solution for all asthma cases by comparing the following information criteria: Akaike’s information criterion (AIC), adjusted Bayesian Information Criterion (aBIC), and interpretability, and chose the four-class model. We tested if there is a need for subgroup analyses by comparing the four-class model with the variance locked among men and women to a freely estimated four-class model. The fit of the freely estimated model was found to be significantly better, which suggested that the LCA should be conducted separately for women and men. Since the methacholine challenge was not performed on all cases, we conducted sensitivity analyses to assess the importance of the missing data. This was done by comparing a model that was based on cases who had methacholine challenge performed on the model among all cases. The biggest alteration in the sensitivity analysis was that the number of cases in the most severe asthma subtypes was reduced. Subjects missing the methacholine challenge were mainly classified by other variables. However, the methacholine challenge results were found to provide valuable additional information in the classification and, therefore, the missing data was also included.

## 3. Results

### 3.1. Characteristics of the Study Population

Table 2 presents the characteristics of the study population. Among cases, 66.5% were women. Altogether, 41 cases (7.9%) were 60 years or older.

### 3.2. Latent Classes

The models with two to four classes among men and three to five classes among women had similar fit-indices (Table 3). For men, the decision concerning the number of classes was easier, as both AIC and aBIC suggested the four-class model. For women, this was more difficult, as different fit-indices suggested different solutions. Because of this, we compared the interpretability of the three-to six-class models. We chose the five-class model, as it showed better stability (i.e., the number of starting values associated with the best fitting model). The models were tested with multiple sets of starting values. Figure 1 displays this decision process.

In women, the subtypes formed in the LCA were 1. *Moderate asthma* (0.16 of the population, 95% CI 0.09 to 0.23), 2. *Cough-variant asthma* (0.31 (0.21 to 0.42)), 3. *Eosinophilic asthma* (0.16 (0.10 to 0.22)), 4. *Allergic asthma* (0.20 (0.11 to 0.30)), and 5. *Difficult asthma* (0.16, 0.11 to 0.22). *Moderate asthma* was characterized by mild obstruction (probability of obstruction 0.92 (0.82 to 1.00)), mild-to-moderate BHR (0.47 (0.27 to 0.67) with moderate BHR), very low blood eosinophil levels (0.96 (0.80 to 1.00) with <0.25 × 10^9^/L), and mostly no allergic sensitization. They also reported the least symptoms. *Cough-variant asthma* was characterized by no significant findings in spirometry, no small-airways obstruction, little BHR, and no sensitization (0.86 (0.71 to 1.00)). Although they showed few clinical findings, 0.85 (0.72–0.98) reported prolonged cough and 0.67 (0.56–0.78) occasional wheezing. The cases in this class had their diagnosis based mainly on significant reversibility in spirometry and/or PEF-follow-up (0.83 (0.73 to 0.93) of the cases in this class). *Eosinophilic asthma* was characterized by very mild-to-mild obstruction, high reversibility in lung function, marked BHR (0.82 (0.62 to 1.00) with marked BHR), some allergic sensitization, and the highest amount of moderate wheezing (0.88 (0.76 to 1.00)). *Allergic asthma* was characterized by normal spirometry, smallest proportion of reversibility (0.65 (0.48 to 0.81 with reversibility)), mild-to-moderate BHR, low eosinophil-levels (0.57 (0.40 to 0.74) with <0.25 × 10^9^/L), and second highest amount of moderate wheezing (0.79 (0.67 to 0.92)). *Difficult asthma* was characterized by severe obstruction (0.67 (0.49 to 0.85)). A total of 0.96 (95% CI 0.91 to 1.00) had reversibility in lung function, and 0.59 (0.43 to 0.75) had low blood eosinophil levels. Almost half (0.49 0.33 to 0.65) had polysensitization. Despite significant clinical findings, these cases did not experience the most symptoms.

In men, the four subtypes identified were 1. *Mild asthma* (0.33 (95% CI 0.22 to 0.43)), 2. *Moderate asthma* (0.35 (0.27 to 0.44)), 3. *Allergic asthma* (0.22 (0.13 to 0.31)), and 4. *Difficult asthma* (0.10 (0.05 to 0.15). Men with *Mild asthma* had only a few findings in spirometry (0.60 (0.44 to 0.76) with no bronchial dysfunction), no BHR (0.71 (0.49 to 0.92), and an almost equal probability for low and high blood eosinophilia. *Moderate asthma* was characterized by moderate-to-severe obstruction (0.31 (0.18 to 0.43) with severe obstruction), high probability for small airway obstruction (0.95 (0.85 to 1.00)), moderate-to-marked BHR, almost half showing polysensitization, and relatively low symptom probabilities (0.32 (0.19 to 0.45) reported prolonged cough and 0.70 (0.57 to 0.83) moderate wheezing). *Allergic asthma* was similar with the corresponding subtype in women, with the exception that 0.56 (0.35 to 0.77) had high blood eosinophil levels. *Difficult asthma* differed from the corresponding subtype in women in that their obstruction was in 0.82 (0.59 to 1.00) very severe, and 0.64 (0.39 to 0.90) showed both obstruction and restriction. In this class, small airways dysfunction was high, BHR ranged widely, and 0.43 (0.17 to 0.69) had polysensitization. Despite distinct findings in clinical measurements, these cases reported the least symptoms.

Three of the subtypes were similar among women and men: *Allergic, Moderate*, and *Difficult* asthma. *Cough-variant* and *Eosinophilic asthma* were detected only among women, and *Mild asthma* was detected only among men. The latent classes and their item-response probabilities are displayed in Table 4.

We calculated the number of cases with a posterior probability of less than 0.50 of belonging to their best fitting subtype. Among women, 27 (5.2%) subjects showed a poor fit for their best fitting class: eight subjects (13.8%) for *Moderate asthma*, seven subjects (6.4%) for *Cough-variant asthma*, six subjects (10.7%) for *Eosinophilic asthma*, zero for *Allergic asthma,* and six subjects (11.3%) for *Difficult asthma*. Among men, only three (0.6%) cases showed poor class fit, and all of these belonged to the *Moderate asthma* (4.7% of the class).

### 3.3. Factors Predicting Latent Class Membership

We identified the risk factor profiles for the subtypes of adult-onset asthma by fitting the following determinants in multivariate regression analyses: age (categorized as <35 years, 35–49 years, and ≥50 years), body mass index (BMI) (≤25, >25–30, >30), smoking (never, former, current smoker), and parental asthma (no, maternal asthma only, paternal asthma only, both parents having asthma). The analyses were conducted separately among women and men. The risk ratios for risk factors for the different subtypes of asthma with all the other subtypes forming the reference category are displayed in Table 5, and the risk factor profiles are presented in Figure 2.

For *Mild asthma,* no statistically significant predictive factors were identified but BMI >25–30, BMI > 30, former smoking, paternal asthma, and both parents having asthma were found to be related to increased RR (Table 5).

The RR for *Moderate asthma* was statistically significantly and increased among women in relation to age > 50 years (RR 2.59 (1.21 to 5.54)) and former smoking (RR 2.21 (1.19 to 4.11)). Among men, age > 50 years was related to significantly increased RR for developing *Moderate asthma* (RR 1.88 (1.06 to 3.30)), while age 35–49, BMI > 25–30, current smoking, and paternal asthma showed slightly increased RR (Table 5).

The cases with *Allergic asthma* among women were predominantly younger and non-smokers. The RR for >50 years of age was 0.34 (0.18 to 0.67), and the RRs for former and current smoking were also below 1, although not statistically significant (Table 5). Among women, the RR in relation to both parents having asthma was increased for *Allergic asthma*, while maternal and paternal asthma alone did not show any relation with *Allergic asthma*. Among men, RR in relation to increasing age (RR for >50 years 0.34 (0.15 to 0.78)), BMI 25–30, and former and current smoking were also below 1 for *Allergic asthma* (Table 5). Maternal asthma showed increased, but nonsignificant RR for *Allergic asthma* among men (RR 1.68 (0.75 to 3.77)).

The RR for developing *Eosinophilic asthma* was decreased in relation to increased age (RR > 50 years 0.56 (0.28 to 1.13)). Both parents having asthma was a significant risk factor for this subtype of asthma (RR 3.55 (1.09 to 11.62), and the point estimates of RR were increased for maternal and paternal asthma, increasing BMI, and former and current smoking (Table 5).

Older age predicted the risk of *Cough-variant asthma* (RR for age > 50 years 1.65 (0.99–2.78)) as did BMI > 25–30 (RR 1.35 (0.93 to 1.97)), while current and former smoking did not predict this subtype. Heredity did not show any relation with the onset of *Cough-variant asthma* (Table 5).

For *Difficult asthma* among women, the RR was increased in relation to age > 50 years, BMI > 30 (RR 1.37 (0.69 to 2.70)), and current smoking (RR 1.25 (0.65 to 2.40)) (Table 5). Heredity did not show any relation with onset of this subtype. Among men, the RR of developing *Difficult asthma* was increased in relation to age > 50 years (RR 1.28 (0.39 to 4.22), former smoking (RR 6.63 (0.87 to 50.42)), current smoking (RR 7.56 (0.98 to 58.21)), and maternal asthma (RR 2.52 (0.72 to 8.83).

We also estimated the RR for having ACOS in relation to the severity of the chronic airways obstruction. Among women, altogether 3 cases (7.5% of the class) among those being originally categorized as *Moderate asthma* and 5 cases (12.2%) as *Difficult asthma* were found to fulfil the diagnostic criteria for ACOS, which is a disease entity identified after the original FEAS was conducted [22]. The risk ratio of ACOS predicting *Moderate asthma* in women was 2.48 (0.97 to 6.37) and *Difficult asthma* in women 4.25 (2.30 to 7.87). In men, 8 cases (16.0%) in the *Moderate asthma* class and 9 cases (64.3) in the *Difficult asthma* class fulfilled the diagnostic criteria for ACOS. The risk ratio (95% CI) was 1.34 (0.77 to 2.35) for *Moderate asthma* and 12.7 (4.83 to 33.45) for *Difficult asthma*.

## 4. Discussion

In the present study, we formed subtypes of Aduat-onset asthma by applying several clinical measurements and the symptoms reported at the time of diagnosis. This is, to date, the largest study that has been conducted among adult-onset asthmatics, where subtypes of asthma were formed with unsupervised clustering methods. This study shows, for the first time, that subtypes of adult-onset asthma differ between men and women. Another main finding of the present study was that the risk factor profiles differ substantially between the different asthma subtypes that were identified. We were able to include all of the major clinical measurements that are usually measured in the diagnosing stage for asthma.

### 4.1. Validity of Results

The FEAS-study is a population-based, incident case-control study during which we recruited all new cases of adult-onset asthma which were diagnosed during the study period in the geographically defined study area in Southern Finland. We achieved a good response rate of 90% through the healthcare system and 78% through the National Social Insurance Institution. Since there is a strict policy by which asthma medications are reimbursed after the initial asthma diagnosis, the patients of this study are most likely asthmatics. Therefore, the subtypes formed in this study are likely to represent the adult-onset asthma subtypes in the general northern European population. We were able to include the major clinical measurements that are usually used in the diagnosing stage for asthma. This makes our subtyping relevant for outpatient clinics conducting such subtyping at an early stage of asthma.

At the time of the FEAS data collection, a methacholine challenge was performed only among those study subjects whose diagnosis remained unclear based on other lung function tests. We conducted sensitivity analyses concerning potential influence of fewer methacholine results. LCA allows for missing information in the classification variables and assumes that the information is missing completely at random [16]. This potentially creates bias, since the methacholine challenge cannot be performed on most severe asthma cases and is not required if an asthma diagnosis can be confirmed by other tests. However, we conducted sensitivity analyses and noticed that the cases with missing data were mostly classified by other variables.

Since we applied LCA, we formed categories for variables that were originally continuous. The cut-offs were chosen to reflect clinical decision-making. As an example, for blood eosinophil levels, we also tested the threshold of 0.30 × 10^9^/L and a three-category variable with the limits <0.17 × 10^9^/L and ≥0.47 × 10^9^/L, and the results were unaltered [27]. For spirometry, we implemented the cut-offs for classifying the severity of obstruction according to the latest standards [28]. Since we used our own controls as the reference category, we excluded from this reference population all smokers and, thus, calculated the spirometry z-values more accurately.

We noticed that adding gender as a classification variable or a risk factor was not enough to account for the differences between men and women. This was compatible with the results of our previous study investigating asthma subtypes based on asthma severity and control [10]. Because of the higher incidence of adult-onset asthma among women, the subtypes formed among men and women combined reflected more than those detectable in women, so relevant subtypes among men would have been missed. However, conducting the analyses separately for women and men caused the data in the subcategories to be smaller, which was seen in the lower number of starting values with the best model (i.e., existence of local maxima), and in the analyses identifying risk factors for different subtypes. On the other hand, when more specific subtypes were formed, the risk factor analyses were more specific providing more useful information for clinical and public health applications. The existence of a global maximum was assured by iterating the model.

For the analyses of the factors predicting the various latent classes, we inserted the cases in their best fitting latent class and calculated risk ratios using Poisson regression. In the LCA context this is not optimal, since it does not allow for the predictive variables to change the posterior probabilities for latent class membership [16]. However, in a population relatively small for LCA, it increased the power and, therefore, decreased the potential effect of chance.

### 4.2. Synthesis with Previous Knowledge

Our analyses differ from other studies on this topic in that our population consisted of subjects with new adult-onset asthma. Thus, our adult-onset asthma cases had not received any asthma treatment before, apart from a potential salbutamol inhaler as needed. In addition, we used only asthma manifestation variables in the classification. Therefore, the results cannot be directly compared with other studies that have used also variables related to asthma treatment. Based on our systematic literature search, only one previous study using clinical data and our own previous study using questionnaire data have formed subtypes of adult asthma separately for women and men [10,11].

As mentioned, there are two previous studies on subtyping adult-onset asthma. In the study of Amelink et al., the suggested subtypes were: (1) Severe eosinophilic inflammation predominant, (2) Frequent symptoms, high healthcare utilization and low sputum eosinophils, (3) mild-to-moderate, well controlled asthma [3]. In the study of Ilmarinen et al., they were: (1) Nonrhinitic controlled to partially controlled asthma with low use of medication or health care, (2) Smoking asthma or ACOS with poor lung function, high symptoms, and high use of medication and health care, (3) Female asthma with normal clinical parameters but relatively high use of health care, (4) Obesity-related asthma with comorbidities, high symptoms and high use of medication; and (5) Atopic well-controlled asthma with onset earlier in adulthood [4].

Some similar phenomena were identified between the present results and previous studies. In the study by Ilmarinen et al., their atopic cluster showed male predominance as well as a good lung function levels [4]. In our study, those with *Allergic asthma* had good lung function. Amelink et al., presented a cluster which consisted mostly of obese women with abundant symptoms [3]. In our study, obesity was mostly related with the subtypes where clinical manifestations were more severe. A subtype close to our *Cough-variant asthma* was found in a study on a general adult population with asthma by Siroux et al. [29]. They found, mainly in women, a subtype characterized by chronic cough and adult-onset . We detected in our study that the probability of having respiratory symptoms was not the highest in the most severe asthma subtypes. This finding is compatible with our previous study that described subtypes of asthma based on asthma control and severity conducted on the Northern Finnish Asthma Study population [10]. Among women, we found an eosinophilic asthma subtype which is compatible with a subtype characterized in literature previously [30]. Furthermore, we detected that eosinophilic inflammation can be present in both allergic and non-allergic subtypes, which has been suggested in a recent review [31].

## 5. Conclusions

The present population-based study of adult-onset asthma was able to identify clinically meaningful subtypes on a newly-diagnosed adult-onset asthma population. Among women, five subtypes of adult-onset asthma were formed: *Moderate asthma*, *Cough-variant asthma*, *Eosinophilic asthma*, *Allergic asthma*, and *Difficult asthma.* Among men, four subtypes of adult-onset asthma were identified: *Mild asthma, Moderate asthma, Allergic asthma,* and *Difficult asthma*. The present study is, to our knowledge, the first study that identified subtypes of adult-onset asthma separately for women and men. We were also able to identify different risk factor profiles for the subtypes identified. The identification of different subtypes with different risk factor profiles opens possibilities for more effective preventive actions in public health practice, and the development of subtype-specific treatment and management of adult-onset asthma in the clinical setting. Since we do not yet have a follow-up on the asthma cases, we were not able to address the question of the best suitable medication for each subtype formed, or how the asthmatics move from one subtype to another in time. This calls for further studies. However, we found that the subtypes formed in the present study fit the population well and were easy to interpret.

## Figures and Tables

**Figure 1 ijerph-20-03072-f001:**
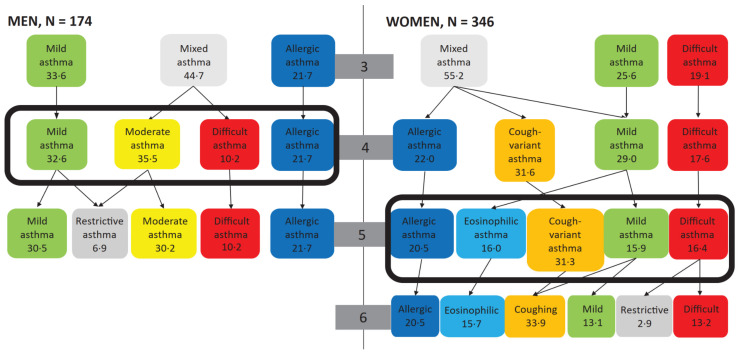
**Suggested subtypes with different numbers of classes chosen.** In the Latent Class Analyses among men and women separately, the fitness indices did not clearly show which number of classes fit the data best. Therefore, we applied the interpretability of the formed classes in clinical practice to support this decision making. Among men, solutions with two to four classes, and in women, solutions with three to five classes, had similar fit-indices. In both genders, the three-class solution produced a class with no clinical interpretability, labeled here as mixed asthma. In men, the following four-class solution produced the most clinically relevant subtypes: 1. *Mild asthma*, 2. *Moderate asthma*, 3. *Difficult asthma*, and 4. *Allergic asthma*. In women, the most meaningful solution was the five-class model: 1. *Mild Asthma*, 2. *Cough-variant asthma*, 3. *Eosinophilic asthma*, 4. *Allergic asthma*, and 5. *Difficult Asthma*. The arrows indicate which classes are divided into further classes, although this is not as straightforward in LCA. The light grey classes are those with no clear clinical interpretability, or classes approaching zero.

**Figure 2 ijerph-20-03072-f002:**
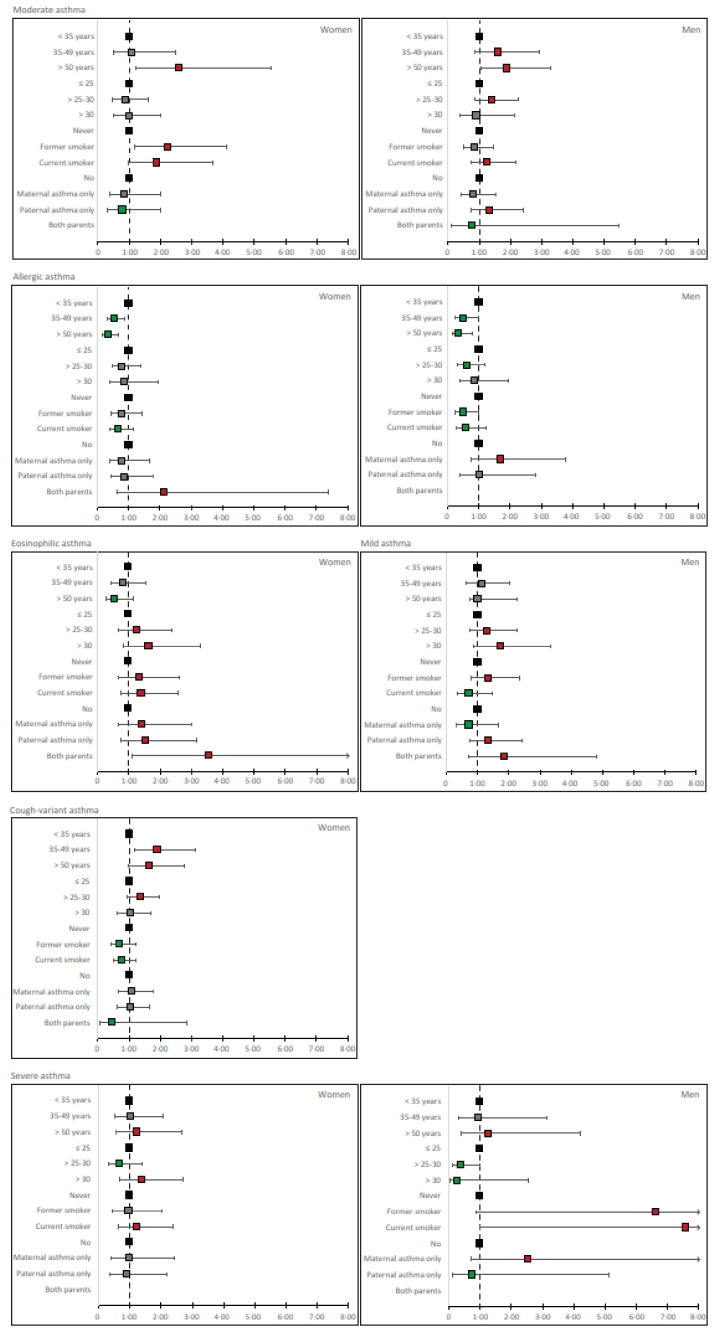
**Risk factor profiles of the formed subtypes.** The risk ratios and their 95% confidence intervals are presented for each subtype identified in forest plots on the natural logarithmic scale. For men and women, the corresponding subtypes of asthma are displayed in the same column and the last column presents the subtypes with no correspondence in the other sex. Risk factors that show risk estimates well above one are presented in red, those with risk estimates well below one are presented in green, and those that show no impact are marked as grey. The black squares represent the reference categories. Increasing age predicts the risk of *Moderate* and *Difficult asthma* among both genders, and *Cough-variant asthma* among women. Age was related to decreased risk of *Allergic asthma* and *Eosinophilic asthma,* suggesting that these subtypes onset only during the younger age period. High BMI showed an increased risk ratio for *Mild* and *Moderate asthma* in men and for *Cough-variant, Eosinophilic* and *Difficult asthma* in women, while it showed decreased risk ratios for *Allergic asthma* among men and women and *Difficult asthma* among men. Former smoking predicted *Mild* and *Difficult asthma* in men, and *Moderate* and *Eosinophilic asthma* in women. Current smoking predicted *Moderate* and *Difficult asthma* among both genders, and *Eosinophilic asthma* in women. In contrast current smoking showed decreased RRs for *Allergic* and *Cough-variant asthma* suggesting selective health behavior among those with allergies. The effects of heredity of asthma depended on the parent in question, but the highest point estimates were seen for *Eosinophilic asthma* among women, *Allergic asthma* among both genders and *Difficult asthma* among men.

**Table 1 ijerph-20-03072-t001:** Clinical measurements among the asthma cases.

	Total Case Population, N (%)	Women, N (%)	Men, N (%)
	520 (100)	346 (66.5)	174 (33.5)
**Type of ventilatory dysfunction in spirometry according to FEV1/FVC and FVC ***			
No obstruction or restriction (FEV1/FVC and FVC z-value ≥ −1.65)	210 (43.7)	148 (47.0)	62 (37.4)
Obstruction (FEV1/FVC < −1.65 and FVC ≥ −1.65)	223 (46.4)	135 (42.9)	88 (53.0)
Restriction (FEV1/FVC ≥ −1.65 and FVC < −1.65)	16 (3.3)	11 (3.5)	5 (3.0)
Obstruction and restriction (FEV1/FVC and FVC < −1.65)	32 (6.7)	21 (6.7)	11 (6.6)
**Severity of obstruction according to FEV1**			
Very mild (FEV1 z-value > −1.65)	335 (69.4)	228 (72.2)	107 (64.1)
Mild (−1.65–−1.99)	36 (7.5)	24 (7.6)	12 (7.2)
Moderate (−2.00–−2.49)	38 (7.9)	25 (7.9)	13 (7.8)
Severe (−2.50–−3.99)	55 (11.4)	34 (10.8)	21 (12.6)
Very severe (≤−4.00)	19 (3.9)	5 (1.6)	14 (8.4)
**Small airways obstruction**			
No (FEF50 z-value > −1.65)	267 (56.6)	178 (58.2)	89 (53.6)
Yes (≤−1.65)	205 (43.4)	128 (41.8)	77 (46.4)
**ACOS**			
No	365 (93.6)	245 (96.8)	120 (87.6)
Yes	25 (6.4)	8 (3.2)	17 (12.4)
**Reversibility in Spirometry or PEF follow-up**			
No	77 (15.9)	54 (16.7)	23 (14.5)
Yes	406 (84.1)	270 (83.3)	136 (85.5)
**Bronchial hyperresponsiveness in the metacholine challenge**			
No	82 (26.3)	52 (23.9)	30 (30.9)
Mild	95 (30.5)	69 (31.7)	26 (27.7)
Moderate	75 (24.0)	54 (24.8)	21 (22.3)
Marked	60 (19.2)	43 (19.7)	17 (18.1)
**Blood eosinophil level**			
<0.25 × 10^9^/L	246 (60.2)	174 (62.1)	72 (55.8)
≥0.25 × 10^9^/L	163 (39.9)	106 (37.9)	57 (44.2)
**Allergic sensitization in Phadiatop**			
No	266 (55.0)	187 (57.4)	79 (50.0)
Sensitized	21 (4.3)	15 (4.6)	6 (3.8)
Polysensitised	197 (40.7)	124 (38.0)	73 (46.2)
**Persistent cough for over 2 months**			
No	250 (48.1)	137 (39.6)	113 (64.9)
Yes	270 (51.9)	209 (60.4)	61 (35.1)
**Nocturnal cough, wheezing or shortness of breath**			
No	187 (36.0)	109 (31.5)	78 (44.8)
Yes	333 (64.0)	237 (68.5)	96 (55.2)
**Occasional wheezing**			
No	139 (26.7)	93 (26.9)	46 (26.4)
Yes	381 (73.3)	253 (73.1)	128 (73.6)
**Attacks of shortness of breath**			
No	78 (15.0)	50 (14.5)	28 (16.1)
Yes	442 (85.0)	296 (85.6)	146 (83.9)
**Exercise-induced shortness of breath**			
No	121 (23.3)	85 (24.6)	36 (20.7)
Yes	399 (76.7)	261 (75.4)	138 (79.3)

* The z-values were calculated from the residual standard deviation of the control population applying linear regression. **Missing information:** Ventilatory dysfunction 39, 31, 8; FEV1 37, 30, 8; Small airways obstruction 48, 40, 8; ACOS 130, 93, 37; Reversibility 37, 22, 15; Methacholine Challenge 208, 128, 80; Blood Eosinophil Level 111, 66, 45; Allergic Sensitization in PhadiAtop 36, 20, 16.

**Table 2 ijerph-20-03072-t002:** Characteristics of the asthma cases.

	Total Case Population, N (%)	Women, N (%)	Men, N (%)
	520 (100)	346 (66.5)	174 (33.5)
**Age**			
<35	158 (30.4)	101 (29.2)	57 (32.8)
35–49	181 (34.8)	122 (35.3)	59 (33.9)
≥50	181 (34.8)	123 (35.6)	58 (33.3)
**BMI**			
≤25	217 (44.8)	156 (49.4)	61 (36.3)
>25 to ≤30	179 (37.0)	100 (31.7)	79 (47.0)
>30	88 (18.2)	60 (19.0)	28 (16.7)
**Smoking**			
Never	238 (46.0)	183 (53.2)	55 (31.8)
Ex	133 (25.7)	70 (20.4)	63 (36.4)
Current	146 (28.2)	91 (26.5)	55 (31.8)
**Parental asthma**			
No	356 (74.8)	243 (75.2)	113 (73.9)
Maternal asthma only	58 (12.2)	38 (11.8)	20 (13.1)
Paternal asthma only	54 (11.3)	37 (11.5)	17 (11.1)
Both parents with asthma	8 (1.7)	5 (1.6)	3 (2.0)

Missing values N total population, women, men: BMI 36, 30, 6; Smoking 3, 2, 1; Parental Asthma 44, 23, 21.

**Table 3 ijerph-20-03072-t003:** Information criteria for the different class solutions for women and men.

Number of Classes	1	2	3	4	5	6
**Women**						
Log likelihood	−2158.62	−1983.41	−1952.19	−1930.47	−1909.08	−1890.98
G-squared	1138.93	788.51	726.08	682.64	639.85	603.66
AIC	1172.93	858.51	832.08	824.64	817.85	**817.66**
BIC	1238.32	**993.14**	1035.94	1097.74	1160.18	1229.23
CAIC	1255.32	**1028.14**	1088.94	1168.74	1249.18	1336.23
Adjusted BIC	1184.39	882.11	**867.81**	872.50	877.85	889.80
Entropy	1.00	0.80	0.81	0.73	0.74	0.83
DF	7662	7644	7626	7608	7590	7572
Percentage of starting values associated with the best model *	100.0	100.0	15.5	20.4	32.7	0.4
**Men**						
Log likelihood	−1096.02	−994.97	−974.81	−955.89	−942.55	−929.22
G-squared	669.82	467.72	427.39	389.55	362.86	336.21
AIC	703.82	537.72	533.39	**531.55**	540.86	550.21
BIC	757.52	**648.29**	700.82	755.84	822.02	888.23
CAIC	774.52	**683.29**	753.82	826.84	911.02	995.23
Adjusted BIC	703.69	537.45	532.98	**531.01**	540.19	549.40
Entropy	1.00	0.88	0.81	0.82	0.84	0.84
DF	7662	7644	7626	7608	7590	7572
No. of starting values associated with the best model *	100.0	100.0	54.7	23.1	4.6	0.4

The bolded number indicates the best fitting solution according to the information criteria analyzed. AIC = Akaike’s Information Criterion, BIC = Bayesian Information Criterion, CAIC = Corrected Akaike’s Information Criterion, Adjusted BIC = Adjusted Bayesian Information Criterion, DF = Degrees of Freedom. * Indicates the number of starting values of 1000 iterations associated with the best fitting model.

**Table 4 ijerph-20-03072-t004:** Latent classes and the corresponding item-response probabilities for women and men.

	Women, N = 346					Men, N = 174			
	**Women** * **Moderate asthma** *	**Women** * **Cough-variant asthma** *	**Women** * **Eosinophilic asthma** *	**Women** * **Allergic asthma** *	**Women** * **Difficult asthma** *	**Men** * **Mild asthma** *	**Men** * **Moderate asthma** *	**Men** * **Allergic asthma** *	**Men** * **Difficult asthma** *
Class membership probabilities	0.16 (0.09–0.23)	0.31 (0.21–0.42)	0.16 (0.10–0.22)	0.20 (0.11–0.30)	0.16 (0.11–0.22)	0.33 (0.22–0.43)	0.35 (0.27–0.44)	0.22 (0.13–0.31)	0.10 (0.05–0.15)
	**Item response probabilities**								
	**Type of ventilatory dysfunction**								
None	0.01 (0.00–0.07)	**0.82 (0.69–0.94)**	0.09 (0.00–0.19)	**0.93 (0.81–1.00)**	0.00 (0.00–0.02)	**0.60 (0.44–0.76)**	0.00 (0.00–0.02)	**0.81 (0.66–0.96)**	0.01 (0.00–0.04)
Obstruction	**0.92 (0.82–1.00)**	0.18 (0.06–0.31)	**0.90 (0.79–1.00)**	0.07 (0.00–0.19)	**0.43 (0.28–0.58)**	0.36 (0.21–0.52)	**0.94 (0.88–1.00)**	0.19 (0.04–0.34)	0.35 (0.10–0.60)
Restriction	0.06 (0.00–0.15)	0.00 (0.00–0.07)	0.00 (0.00–0.07)	0.00 (0.00–0.00)	0.15 (0.03–0.28)	0.04 (0.00–0.09)	0.05 (0.00–0.11)	0.00 (0.00–0.01)	0.00 (0.00–0.01)
Obstruction and restriction	0.00 (0.00–0.01)	0.00 (0.00–0.01)	0.00 (0.00–0.01)	0.00 (0.00–0.00)	**0.42 (0.26–0.58)**	0.00 (0.00–0.01)	0.00 (0.00–0.01)	0.00 (0.00–0.01)	**0.64 (0.39–0.90)**
	**Severity of obstruction**								
Very mild	**0.61 (0.41–0.81)**	**0.98 (0.95–1.00)**	**0.67 (0.50–0.84)**	**1.00 (0.99–1.00)**	0.00 (0.00–0.03)	**0.92 (0.83–1.00)**	0.37 (0.24–0.51)	**0.94 (0.86–1.00)**	0.01 (0.00–0.09)
Mild	0.25 (0.09–0.41)	0.02 (0.00–0.05)	0.19 (0.05–0.33)	0.00 (0.00–0.01)	0.00 (0.00–0.01)	0.06 (0.00–0.14)	0.12 (0.03–0.21)	0.05 (0.00–0.13)	0.00 (0.00–0.03)
Moderate	0.14 (0.00–0.29)	0.00 (0.00–0.00)	0.13 (0.00–0.29)	0.00 (0.00–0.00)	0.22 (0.04–0.41)	0.02 (0.00–0.06)	**0.20 (0.09–0.31)**	0.00 (0.00–0.01)	0.00 (0.00–0.03)
Severe	0.00 (0.00–0.02)	0.00 (0.00–0.02)	0.00 (0.00–0.02)	0.00 (0.00–0.00)	**0.67 (0.49–0.85)**	0.00 (0.00–0.01)	**0.31 (0.18–0.43)**	0.00 (0.00–0.01)	0.17 (0.00–0.38)
Very severe	0.00 (0.00–0.00)	0.00 (0.00–0.00)	0.00 (0.00–0.00)	0.00 (0.00–0.00)	0.10 (0.01–0.19)	0.00 (0.00–0.01)	0.00 (0.00–0.02)	0.00 (0.00–0.01)	**0.82 (0.59–1.00)**
	**Small airways dysfunction**								
No	0.23 (0.03–0.43)	**0.88 (0.80–0.97)**	0.27 (0.10–0.43)	**0.98 (0.92–1.00)**	0.05 (0.00–0.15)	**0.91 (0.81–1.00)**	0.05 (0.00–0.15)	**1.00 (0.97–1.00)**	0.01 (0.00–0.05)
Yes	**0.77 (0.57–0.97)**	0.12 (0.03–0.20)	**0.73 (0.57–0.90)**	0.02 (0.00–0.08)	**0.95 (0.85–1.00)**	0.09 (0.00–0.19)	**0.95 (0.85–1.00)**	0.00 (0.00–0.03)	**0.99 (0.95–1.00)**
	**Reversibility in lung function**								
No	0.19 (0.05–0.32)	0.17 (0.07–0.27)	0.05 (0.00–0.12)	0.35 (0.19–0.52)	0.04 (0.00–0.09)	0.05 (0.00–0.13)	0.05 (0.00–0.11)	**0.56 (0.34–0.78)**	0.00 (0.00–0.02)
Yes	**0.81 (0.68–0.95)**	**0.83 (0.73–0.93)**	**0.95 (0.88–1.00)**	**0.65 (0.48–0.81)**	**0.96 (0.91–1.00)**	**0.95 (0.87–1.00)**	**0.95 (0.89–1.00)**	0.44 (0.22–0.66)	**1.00 (0.98–1.00)**
	**Bronchial hyperresponsiveness**								
No	0.07 (0.00–0.17)	**0.49 (0.34–0.64)**	0.01 (0.00–0.06)	0.19 (0.04–0.35)	0.05 (0.00–0.14)	**0.71 (0.49–0.92)**	0.11 (0.00–0.25)	0.08 (0.00–0.24)	0.03 (0.00–0.29)
Mild	**0.43 (0.23–0.63)**	**0.41 (0.27–0.54)**	0.01 (0.00–0.06)	**0.31 (0.15–0.46)**	0.25 (0.04–0.47)	0.12 (0.00–0.27)	0.26 (0.08–0.43)	**0.47 (0.26–0.68)**	**0.48 (0.00–1.00)**
Moderate	**0.47 (0.27–0.67)**	0.10 (0.00–0.23)	0.17 (0.00–0.36)	**0.34 (0.17–0.50)**	0.29 (0.07–0.50)	0.07 (0.00–0.18)	**0.33 (0.14–0.51)**	**0.32 (0.13–0.52)**	0.03 (0.00–0.25)
Marked	0.03 (0.00–0.13)	0.00 (0.00–0.01)	**0.82 (0.62–1.00)**	0.17 (0.04–0.29)	**0.41 (0.17–0.65)**	0.11 (0.00–0.24)	**0.31 (0.13–0.49)**	0.13 (0.00–0.27)	**0.46 (0.00–1.00)**
	**Blood eosinophil level**								
<0.25 × 10^9^/L	**0.96 (0.80–1.00)**	**0.75 (0.63–0.87)**	0.10 (0.00–0.35)	**0.57 (0.40–0.74)**	**0.59 (0.43–0.75)**	**0.58 (0.42–0.75)**	**0.62 (0.46–0.77)**	0.44 (0.23–0.65)	**0.52 (0.17–0.87)**
≥0.25 × 10^9^/L	0.04 (0.00–0.20)	0.25 (0.13–0.37)	**0.90 (0.65–1.00)**	0.43 (0.26–0.60)	0.41 (0.25–0.57)	0.42 (0.25–0.58)	0.38 (0.23–0.54)	**0.56 (0.35–0.77)**	0.48 (0.13–0.83)
	**Allergic sensitization**								
No	**0.74 (0.57–0.91)**	**0.86 (0.71–1.00)**	**0.43 (0.27–0.60)**	0.20 (0.00–0.41)	**0.46 (0.31–0.62)**	**0.76 (0.61–0.91)**	**0.45 (0.30–0.60)**	0.14 (0.00–0.33)	**0.57 (0.31–0.83)**
Monosensitization	0.03 (0.00–0.11)	0.03 (0.00–0.09)	0.12 (0.01–0.23)	0.02 (0.00–0.10)	0.05 (0.00–0.11)	0.05 (0.00–0.12)	0.06 (0.00–0.12)	0.00 (0.00–0.01)	0.00 (0.00–0.02)
Polysensitization	0.24 (0.09–0.39)	0.11 (0.00–0.25)	**0.45 (0.28–0.62)**	**0.78 (0.55–1.00)**	**0.49 (0.33–0.65)**	0.19 (0.04–0.33)	**0.49 (0.34–0.64)**	**0.86 (0.66–1.00)**	0.43 (0.17–0.69)
	**Prolonged cough**								
No	**0.55 (0.38–0.73)**	0.15 (0.02–0.28)	0.44 (0.28–0.60)	**0.61 (0.42–0.80)**	0.40 (0.25–0.54)	**0.52 (0.36–0.67)**	**0.68 (0.55–0.81)**	**0.82 (0.67–0.97)**	**0.62 (0.38–0.86)**
Yes	0.45 (0.27–0.62)	**0.85 (0.72–0.98)**	**0.56 (0.40–0.72)**	0.39 (0.21–0.58)	**0.60 (0.46–0.75)**	0.48 (0.33–0.64)	0.32 (0.19–0.45)	0.18 (0.03–0.33)	0.38 (0.14–0.63)
	**Moderate wheezing**								
No	0.38 (0.21–0.55)	0.33 (0.05–0.44)	0.12 (0.00–0.24)	0.21 (0.08–0.33)	0.28 (0.15–0.41)	0.22 (0.08–0.35)	0.30 (0.17–0.43)	0.22 (0.06–0.38)	0.38 (0.14–0.62)
Yes	**0.62 (0.45–0.79)**	**0.67 (0.56–0.78)**	**0.88 (0.76–1.00)**	**0.79 (0.67–0.92)**	**0.72 (0.59–0.85)**	**0.78 (0.65–0.92)**	**0.70 (0.57–0.83)**	**0.78 (0.62–0.94)**	**0.62 (0.38–0.86)**

The bolded number indicates the highest item response probability.

**Table 5 ijerph-20-03072-t005:** Distribution of asthma cases according to their best fitting asthma subtype and risk ratios (RR) for potential determinants.

	Men	Women	Men	Women			Men	Women	Men
	** *Mild asthma* **	** *Moderate asthma* **	** *Moderate asthma* **	** *Cough-variant asthma* **	** *Eosinophilic asthma* **	** *Allergic asthma* **	** *Allergic asthma* **	** *Difficult asthma* **	** *Difficult asthma* **
N (%)	57 (32.8)	58 (16.8)	64 (36.8)	110 (31.8)	56 (16.2)	69 (19.9)	37 (21.3)	53 (15.3)	16 (9.2)
Age									
<35 years	16 (28.1)	12 (20.7)	15 (23.4)	21 (19.1)	19 (33.9)	33 (47.8)	22 (59.5)	16 (30.2)	4
35-49 years	21 (36.8)	13 (22.4)	24 (37.5)	46 (41.8)	21 (37.5)	25 (36.2)	9 (24.3)	17 (32.1)	5
RR (95% CI)	1.12 (0.62–2.02)	1.10 (0.49–2.50)	1.59 (0.86–2.92)	**1.90 (1.16–3.11)**	0.81 (0.44–1.52)	**0.54 (0.32–0.90)**	0.49 (0.24–1.01)	1.03 (0.50–2.09)	0.97 (0.30–3.15)
≥50 years	20 (35.1)	33 (56.9)	25 (39.1)	43 (39.1)	16 (28.6)	11 (15.9)	6 (16.2)	20 (37.7)	7
RR (95% CI)	1.00 (0.73–2.26)	**2.59 (1.21–5.54)**	**1.88 (1.06–3.30)**	1.65 (0.99–2.78)	0.56 (0.28–1.13)	**0.34 (0.18–0.67)**	**0.34 (0.15–0.78)**	1.24 (0.57–2.67)	1.28 (0.39–4.22)
BMI									
≤25	17 (35.1)	25 (48.1)	19 (31.7)	43 (42.2)	22 (44.0)	44 (66.7)	17 (46.0)	22 (47.8)	8 (50.0)
>25-30	26 (47.3)	17 (32.7)	33 (55.0)	39 (38.2)	17 (34.0)	17 (25.8)	14 (37.8)	10 (21.7)	6 (37.5)
RR (95% CI)	1.29 (0.73–2.26)	0.87 (0.47–1.62)	1.41 (0.87–2.27)	1.35 (0.93–1.97)	1.27 (0.67–2.39)	0.81 (0.48–1.37)	0.62 (0.32–1.20)	0.65 (0.30–1.39)	0.37 (0.14–1.01)
>30	12 (21.8)	10 (19.2)	8 (13.3)	20 (19.6)	11 (22.0)	5 (7.6)	6 (16.2)	14 (30.4)	2 (12.5)
RR (95% CI)	1.73 (0.89–3.34)	0.98 (0.49–1.99)	0.92 (0.40–2.12)	1.04 (0.64–1.68)	1.63 (0.81–3.28)	0.42 (0.17–1.06)	0.85 (0.37–1.96)	1.37 (0.69–2.70)	0.28 (0.03–2.52)
Smoking									
Never	17 (30.3)	25 (43.1)	18 (28.1)	64 (58.7)	26 (47.3)	43 (62.3)	19 (51.4)	25 (47.2)	1 (6.3)
Former Smoker	26 (46.4)	17 (29.3)	20 (31.3)	19 (17.4)	12 (21.8)	12 (17.4)	10 (27.0)	10 (18.9)	7 (43.8)
RR (95% CI)	1.34 (0.77–2.33)	**2.21 (1.19–4.11)**	0.83 (0.48–1.44)	0.69 (0.43–1.23)	1.33 (0.68–2.60)	0.80 (0.44–1.44)	0.50 (0.24–1.00)	0.96 (0.46–2.04)	6.63 (0.87–50.42)
Current Smoker	13 (23.2)	16 (27.6)	26 (40.6)	26 (23.9)	17 (30.9)	14 (20.3)	8 (21.6)	18 (34.0)	8 (50.0)
RR (95% CI)	0.72 (0.36–1.45)	1.89 (0.97–3.69)	1.26 (0.73–2.16)	0.79 (0.51–1.23)	1.39 (0.75–2.57)	0.67 (0.39–1.16)	0.58 (0.27–1.25)	1.25 (0.65–2.40)	7.56 (0.98–58.21)
Parental asthma									
No	39 (78.0)	41 (82.0)	40 (71.4)	75 (73.5)	37 (67.3)	50 (75.8)	24 (72.7)	40 (80.0)	10 (71.4)
Maternal asthma only	5 (10.00)	5 (10.0)	7 (12.5)	14 (13.7)	7 (12.7)	7 (10.6)	5 (21.4)	5 (10.0)	3 (21.4)
RR (95% CI)	0.72 (0.31–1.65)	0.85 (0.37–1.99)	0.80 (0.41–1.55)	1.08 (0.66–1.79)	1.42 (0.67–3.00)	0.80 (0.38–1.68)	1.68 (0.75–3.77)	0.99 (0.41–2.43)	2.52 (0.72–8.83)
Paternal asthma only	4 (8.00)	4 (8.00)	8 (14.3)	12 (11.8)	9 (16.4)	7 (10.6)	4 (12.1)	5 (10.0)	1 (7.1)
RR (95% CI)	0.66 (0.29–1.50)	0.78 (0.30–1.99)	1.34 (0.74–2.41)	1.04 (0.64–1.67)	1.54 (0.75–3.17)	0.88 (0.43–1.79)	1.02 (0.37–2.82)	0.91 (0.38–2.19)	0.77 (0.12–5.12)
Both parents	2 (4.00)	0 (0.0)	1 (1.8)	1 (0.98)	2 (3.6)	2 (3.0)	0 (0.0)	0 (0.0)	0 (0.0)
RR (95% CI)	1.85 (0.72–4.80)	NA	0.79 (0.11–5.48)	0.47 (0.08–2.86)	**3.55 (1.09–11.62)**	2.13 (0.61–7.40)	NA	NA	NA

Risk ratios calculated by Poisson regression, while the other classes formed the reference. Bolded ratios statistically significant at *p* < 0.05. Abbreviations: RR = Risk Ratio, 95% CI = 95 percent confidence Interval, NA = not applicable.

## Data Availability

Data are available upon reasonable request, to be made to the corresponding author.

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
