# Peer review of "Subtypes of Adult-Onset Asthma at the Time of Diagnosis: A Latent Class Analysis"

_ijerph, 2023, doi:10.3390/ijerph20043072_

Round 1

Reviewer 1 Report

This paper presents interesting data on clinically meaningful adult-onset asthma subtypes. It will fit out the clinician's perspective in personalized asthma case management, as well as providing useful information for assessing the prognosis.

However, there are some suggestions for the manuscript:

1.      Please be consistent in writing biological name of species (italics) or other specific terms for the entire manuscript, including in the tables or figures explanation

2.      A concise and clear explanation for each table is strongly recommended, since every table or figure should be self-explanatory

3.      In the Methods section (Variables included in the latent class analyses), there is a lot of information which make it complicated to read. Please write it shorter, concise, and systematically (might be in a table or figure to make it easier to read)

4.      Some data in tables and discussion contain a lot of information. Please rewrite the most important information in a simpler sentences to make it easier to understand 

5.      The conclusion should directly reflect the study objectives

Author Response

Reviewer 1

Comments and Suggestions for Authors

This paper presents interesting data on clinically meaningful adult-onset asthma subtypes. It will fit out the clinician's perspective in personalized asthma case management, as well as providing useful information for assessing the prognosis.

However, there are some suggestions for the manuscript:

COMMENT 1.1:  Please be consistent in writing biological name of species (italics) or other specific terms for the entire manuscript, including in the tables or figures explanation

RESPONSE 1.1: We revised and corrected our writing style concerning these issues. As an example, we formatted the subtype-names to italics within the text.

COMMENT 1.2: A concise and clear explanation for each table is strongly recommended, since every table or figure should be self-explanatory

RESPONSE 1.2: We added a heading for the two figures in the article and checked the headlines of the tables in the article.

COMMENT 1.2: In the Methods section (Variables included in the latent class analyses), there is a lot of information which make it complicated to read. Please write it shorter, concise, and systematically (might be in a table or figure to make it easier to read)

RESPONSE 1.2: We have now revised this paragraph to make it easier to read and interpret. We also separated table 1 into two separate tables. Table 1 now presents the variables used in latent class analysis and table 2 presents the demographic characteristics of the asthma cases.

COMMENT 1.3: Some data in tables and discussion contain a lot of information. Please rewrite the most important information in a simpler sentences to make it easier to understand 

RESPONSE 1.3: Due to the type of data analysis (latent class analysis) there are lots of probabilities in the tables including the results. This makes it difficult to present the results in a concise way. However, we have now revised the results section of the text to make it more easily interpretable.

COMMENT 1.3:     The conclusion should directly reflect the study objectives

RESPONSE: We modified the conclusions to better reflect the study objectives.

Reviewer 2 Report

Authors requested to recheck the complete Table 1 Characteristics of the Asthma Cases because lots of mistakes in every subdivision. For example, in table 1 authors has mentioned 520 subjects in the total case population while summing up in the gender (Male and female) basis the 518 was listed.

Authors requested to recheck the complete Table 3 Latent classes and the corresponding item−response probabilities for women and men. Instead of 346 you mentioned 342 women subjects check and correct it.

Author Response

Comments and Suggestions for Authors

COMMENT 2.1: Authors requested to recheck the complete Table 1 Characteristics of the Asthma Cases because lots of mistakes in every subdivision. For example, in table 1 authors has mentioned 520 subjects in the total case population while summing up in the gender (Male and female) basis the 518 was listed.

RESPONSE 2.1: We revised Table 1 and corrected the false number of men and the frequencies of men in each heredity category. Instead of 7 men with missing information on FEV1 there were 8. The other numbers add up, when they are combined with the number of missing data for each variable listed in the footnote of the table.

COMMENT 2.2: Authors requested to recheck the complete Table 3 Latent classes and the corresponding item−response probabilities for women and men. Instead of 346 you mentioned 342 women subjects check and correct it.

RESPONSE 2.2: We have now revised Table 3 (since table 1 was divided, now table 4 in the newer version). We have corrected the number of female cases and checked the probabilities in the table to add up to 100%.

Reviewer 3 Report

The article focuses on identifying subtypes of adult-onset asthma in a group of participants in an existing study, FEAS. The article highlights the different subtypes that were identified and the predictors of the different subtypes.

General comments:

In the introduction, the authors mention studies that have investigating the subtypes of adult asthma but the key findings and how they relate to the current study is not mentioned. The authors need to provide more information about these studies so that the reader is able to contextualise and compare with the current study.

The methods section is so brief, yet it is the core of the study. The authors should address the following;

Clarify on the study design. At some point, there is a mention of controls. Who were the cases and the controls?

Provide more information on the FEAS study beyond providing the reference of a previous publication. What is it about? How are the participants enrolled into the study? What clinical and laboratory measurements done? All these should be brief and aimed at understanding how the current study relates to the FEAS study.

Provide more information on the source of data for the current study. It is not clear whether the study used existing data from FEAS or it is they enrolled participants prospectively as part of the FEAS study.

Provide a description on the key clinical practices that lead to diagnoses. For example, are the diagnoses made using  standardized criteria as part of the research or they are based on a clinicians’ judgement and then data from the clinic records captured by the research team? This is very important, given that these were used to come up with the subtypes of asthma.

The section on data management and analysis can be expanded. There is a lot of information in the results on how the results  in the tables were obtained, which should ideally belong to the data analysis section.

The results section contains a lot of good information but it is difficult to read and follow. It would be good to keep focus on the highlighting the results based on the primary objectives of the study and clearly indicate so in the sub-headings. The tables are very crowded and could be split where possible for ease of reading and understanding. For example,

Table 1 -consider separating into 2 table. One for demographic characteristics (this could also be just presented in the narrative) and second one for clinical and laboratory characteristics.

Similarly, Table 3 shows the class probabilities among men and women, and these are different for each gender. It would have been more meaningful to compare the probabilities in men and women by looking at the same table if the classes/subtypes identified were the same. To reduce the overcrowding and improve readability, it might be better to separate this table into 2 (one for men and one for women).

The results show the different subtypes that were generated by the latent class analysis. These subtypes are not mutually exclusive and could overlap. Some are based on disease severity, others on underlying cellular mechanisms. According to the introduction, the authors hoped that the classes would inform personalized care, and in this case, choice of medication to be used. However, this classification may not achieve that. For example, a person can have severe asthma of the allergic type. The authors need to discuss this issue, which is really a major limitation of the study and its implications in interpretation and use of the study findings. This could have arisen from the diagnoses made.

In addition, there are many terms which are not commonly used in clinical practice based on standard guidelines like GINA and these need to be clearly defined. For example, mixed asthma, restrictive asthma, difficult asthma, coughing, severe wheezing. The authors need to review this classification to ensure that the classes identified are relevant to current knowledge and practice.

Discussion

I note that the discussion focused on the methods used to analyse the data and not the results. The authors need to discuss the results. What were the results, what do they mean in relation to the subject under study, how do they compare with other studies, and what are the implications?

The current information should be discussed as methodological considerations, highlighting the strengths and limitations.

Specific comments

The last paragraph of the introduction (line 53-60) contains information that belongs to the section on data analysis. I would expect that this paragraph clearly stipulates the aims and objectives of the study, and possibly anticipated impact. The type of analysis, the data source and variables analysed should be moved to the relevant sections in methods.

The presentation of risk factors can be improved upon. Since the authors conducted the analyses in men versus women, it may be more logical to follow the same. For example, what were the risk predictors of severe asthma among women and men? This makes it easier to understand and apply clinically.

Author Response

Comments and Suggestions for Authors

The article focuses on identifying subtypes of adult-onset asthma in a group of participants in an existing study, FEAS. The article highlights the different subtypes that were identified and the predictors of the different subtypes.

General comments:

COMMENT 3.1: In the introduction, the authors mention studies that have investigating the subtypes of adult asthma but the key findings and how they relate to the current study is not mentioned. The authors need to provide more information about these studies so that the reader is able to contextualize and compare with the current study.

RESPONSE 3.1: We have now revised the Introduction and added some text on the suggested subtypes formed in these two previous studies (lines 41-44) on adult-onset asthma in the Discussion section (lines 399-408)

COMMENT 3.1: The methods section is so brief, yet it is the core of the study. The authors should address the following;

Clarify on the study design. At some point, there is a mention of controls. Who were the cases and the controls?

RESPONSE 3.1: We added a section concerning the controls in the Methods -section (lines 68-71)

COMMENT 3.2: Provide more information on the FEAS study beyond providing the reference of a previous publication. What is it about? How are the participants enrolled into the study? What clinical and laboratory measurements done? All these should be brief and aimed at understanding how the current study relates to the FEAS study.

RESPONSE 3.2: We elaborated on the study design of the FEAS-study in the Methods section. (lines 64-90)

COMMENT 3.3: Provide more information on the source of data for the current study. It is not clear whether the study used existing data from FEAS or it is they enrolled participants prospectively as part of the FEAS study.

RESPONSE 3.3: The study included all the population-based cases recruited from the source population of the original FEAS study which produced the asthma cases. In the present study the controls were used only for the spirometry reference value calculations. We added these details to the Methods-section.

COMMENT 3.4: Provide a description on the key clinical practices that lead to diagnoses. For example, are the diagnoses made using standardized criteria as part of the research or they are based on a clinicians’ judgement and then data from the clinic records captured by the research team? This is very important, given that these were used to come up with the subtypes of asthma.

RESPONSE 3.4: We have now revised the Methods to provide more information on the diagnostic practices in the FEAS study (see lines 82-91 of the formatted article). Briefly: all new cases of adult-onset asthma were recruited in the study area during the study period, largely through the Tampere University Hospital, but also through the health care centers in the area. They followed criteria that were based on the Finnish Current care guidelines applied at the time of data collection and agreed with the respiratory physicians working at the Tampere University Hospital at the time of data collection. A standardized form explaining these criteria and the pathway of diagnostics was distributed to all physicians who were potentially diagnosing asthma in the study area.  

For the study subjects who were recruited through the National Social Insurance Institution, we checked from their medical records that their diagnosis was comparable with these criteria.

COMMENT 3.4: The section on data management and analysis can be expanded. There is a lot of information in the results on how the results in the tables were obtained, which should ideally belong to the data analysis section.

RESPONSE 3.5: We understand the concern. We moved the section on how we chose the best fitting model into the statistical methods -section (lines 161-167). We also moved the section on sensitivity analyses concerning the missing methacholine challenge test into the statistical methods section (lines 168-175). However, we feel that the rest of the analysis methods are best displayed side-by-side with the corresponding results.

COMMENT 3.5: The results section contains a lot of good information but it is difficult to read and follow. It would be good to keep focus on the highlighting the results based on the primary objectives of the study and clearly indicate so in the sub-headings. The tables are very crowded and could be split where possible for ease of reading and understanding. For example,

RESPONSE 3.5: We have revised the Results -section to make it easier to read. As an example, we have formatted the subtype -names in italic. In addition, we have moved some sections from the Results into the Statistical methods -section

COMMENT 3.6: Table 1 -consider separating into 2 table. One for demographic characteristics (this could also be just presented in the narrative) and second one for clinical and laboratory characteristics.

RESPONSE 3.6: We divided the table into two different tables and shifted them to their individually suitable position